# Morphological and Phylogenetic Analyses Reveal Three New Species of *Phyllosticta* (*Botryosphaeriales*, *Phyllostictaceae*) in China

**DOI:** 10.3390/jof10010007

**Published:** 2023-12-22

**Authors:** Yang Jiang, Zhaoxue Zhang, Jie Zhang, Shi Wang, Xiuguo Zhang

**Affiliations:** 1College of Life Sciences, Shandong Normal University, Jinan 250358, China; jiangyang202309@126.com (Y.J.); wangstone@sdnu.edu.cn (S.W.); 2Shandong Provincial Key Laboratory for Biology of Vegetable Diseases and Insect Pests, College of Plant Protection, Shandong Agricultural University, Taian 271018, China; zhangzhaoxue2022@126.com (Z.Z.); zhjie8087@163.com (J.Z.)

**Keywords:** multigene phylogeny, new species, taxonomy

## Abstract

The genus *Phyllosticta* has been reported worldwide and contains many pathogenic and endophytic species isolated from a wide range of plant hosts. A multipoint phylogeny based on gene coding combinatorial data sets for the internal transcribed spacer (ITS), large subunit of ribosomal RNA (LSU rDNA), translation elongation factor 1α (TEF1α), actin (ACT), and glycerol-3-phosphate dehydrogenase (GPDH), combined with morphological characteristics, was performed. We describe three new species, *P. fujianensis* sp. nov., *P. saprophytica* sp. nov., and *P. turpiniae* sp. nov., and annotate and discusse their similarities and differences in morphological relationships and phylogenetic phases with closely related species.

## 1. Introduction

*Phyllosticta* Pers. was initially established by Persoon in 1818 [1]. At the outset, *Phyllosticta* was categorized within *Botryosphaeriaceae* [2,3]. Later on, it was acknowledged as the anamorph of *Guignardia*, following the recommendations of Viala and Ravaz [4]. The precedence of the earlier name is determined in accordance with the International Code for Nomenclature of Algae, Fungi, and Plants [5]. Slippers et al. [6] placed *Phyllosticta* within *Phyllostictaceae*, *Botryosphaeriales*, relying on phylogenetic relationships. More recently, it has been reclassified in *Botryosphaeriaceae* based on compelling evidence from morphological characteristics and molecular data, particularly concerning conidia covered by mucus. The current classification of *Phyllosticta* poses challenges and warrants reevaluation. *Phyllosticta* is recognized as an endophytic fungi with the capability to induce leaf spot in plants, exhibiting a widespread distribution across the globe. Wang et al., Zhang et al., and van der Aa [7,8,9] have described 46 types of *Phyllosticta*, encompassing 12 sexual forms and 17 spermatial morphs. Van der Aa and Vanev have further modified all *Phyllosticta* species, accepting a total of 190 epithets [10]. The intricate nature of the *Phyllosticta* classification system underscores the need for a thorough reassessment.

*Phyllosticta* represents a highly diverse group, with 3216 scientific names recorded under the genus according to the Index Fungorum search (www.indexfungorum.org, accessed on 25 October 2023) [8]. Identifying *Phyllosticta* poses challenges based solely on morphological characteristics and host combinations, with the host range being broad and unclear. Some *Phyllosticta* species exhibit a wide range of hosts, while others do not. To overcome the limitations associated with morphological characteristics and host combinations, the integration of homologous gene DNA sequencing and comparative methods has significantly advanced our comprehension of the phylogeny of *Phyllosticta* species [5,8,11,12,13,14,15]. Based on ITS, LSU, TEF1 alpha, ACT, and GPDH, five loci of phylogenetic analysis, *Phyllosticta* genetic system development can be divided into six species of composite groups: viz *P. capitalensis* species complex, *P. concentrica* species complex, *P. cruenta* species complex, *P. owaniana* species complex, *P. rhodorae* species complex, and *P. vaccinii* species complex [5,8].

Fujian province is situated in the north latitude range of 23°31′–28°18′ N and the longitude range of 115°50′–120°43′ E. This region experiences an average annual rainfall of 1500 mm, characterized by a subtropical maritime monsoon climate. Fungi were isolated from leaf spots and necrotic leaves of both *Lonicera japonica* and *Turpinia montana* samples.

For the molecular characterization, sequences from five gene loci were employed, including the internal transcribed spacer region of ribosomal DNA (ITS rDNA), the large subunit of ribosomal RNA (LSU rDNA), translation elongation factor 1α (TEF1α), actin (ACT), and glycerol 3-phosphate dehydrogenase (GPDH). Through a combination of phylogenetic and morphological analyses, the fungi were successfully identified as three new species. This discovery adds to the fungal diversity in the region and contributes to understanding of the local ecosystem.

## 2. Materials and Methods

### 2.1. Isolation and Morphology

The leaves of *Lonicera japonica* and *Turpinia montana* Vent, as well as saprophytic leaves collected in Mount Wuyi City, Fujian Province, China, were utilized for this study. Fragments (5 × 5 mm) were retrieved from damaged portions of the samples. Subsequently, the fragments were soaked in 75% alcohol for 1 min, followed by a single rinse with sterile water. Afterwards, they were immersed in a 5% sodium hypochlorite solution for 1 min and rinsed three times with sterile water. The samples were then dried on sterilized filter paper. The samples were inoculated on Potato dextrose agar plate (PDA: 200 g potatoes, 20 g of glucose, 20 g agar, 1000 mL of distilled water, pH 7.0), followed by 2–4 days of incubation at 25 °C. Subsequently, the AGAR portion with fungal mycelium from the periphery of the colony was transferred to a new PDA plate and photographed on days 7 and 15 using a digital camera (Canon Powershot G7X, Canon, Tokyo, Japan). After the appearance of conidia, the microscopic morphological characteristics of the fungi on PDA medium (including conidia, conidial cells, and appendage) were observed using an Olympus SZX10 (OLYMPUS, Tokyo, Japan) stereo microscope and an Olympus BX53 (OLYMPUS, Tokyo, Japan) microscope, and the colony characteristics of the fungi on the PDA medium were recorded. All the devices were equipped with an Olympus DP80 (OLYMPU, Tokyo, Japan) high-definition color digital camera to photograph the fungal structures. All the fungal strains were stored in 10% sterilized glycerine at 4 °C for follow-up research. The holotype specimens were deposited in the Herbarium of Plant Pathology, Shandong Agricultural University (HSAUP) and Herbarium Mycologicum Academiae Sinicae, Institute of Microbiology, Chinese Academy of Sciences, Beijing, China (HMAS). Ex-holotype living cultures were deposited in the Shandong Agricultural University Culture Collection (SAUCC). The taxonomic information of the new taxa was submitted to MycoBank (http://www.mycobank.org, accessed on 25 October 2023).

### 2.2. DNA Extraction and Amplification

The genomic DNA of the fungal mycelium growing on the PDA plate was extracted using a DNA extraction kit (GeneOn BioTech, Ludwigshafen am Rhein, Germany). The ITS, LSU, TEF1α, ACT, and GPDH regions were amplified using the primer pairs and polymerase chain reaction (PCR) programs specified in Table 1. The amplification reactions were conducted in a 20 μL reaction volume, comprising 12.5 µL of 2× Hieff Canace^®^ Plus PCR Master Mix (Yeasen Biotechnology, Shanghai, China, Cat No. 10154ES03), with 1 µL each (10 µM) of forward and reverse primers (TsingKe, Qingdao, China) and 1 µL of template genomic DNA. The volume was adjusted to a total of 25 µL with distilled deionized water. The PCR amplification products were observed on a 2% agarose electrophoresis gel. DNA sequencing was carried out using an Eppendorf Master Thermocycler (Hamburg, Germany) at Tsingke Company Limited (Qingdao, China), bi-directionally. Consistent sequences were obtained using MEGA 7.0. All the sequences generated in this study were deposited in GenBank (Appendix A).

### 2.3. Phylogenetic Analyses

The newly obtained sequencing data were processed using MEGA v.7.0 to ensure sequence consistency. Reference sequences were downloaded based on the GenBank numbers provided in the latest articles [8,13,22,23,24]. For phylogenetic inference, the sequence utilized by Zhang et al. [7] served as the foundation, considering the ITS-LSU-TEF1α-ACT-GAPDH sequence. MEGA v.7.0 software was employed to compare the new sequences from this study with those available in GenBank.

A phylogenetic analysis was conducted utilizing both the maximum likelihood (ML) and Bayesian inference (BI) algorithms. MrModelTest v.2.3 was employed to determine the optimal evolutionary model for each locus region [25], and this information was integrated into the BI analysis. The ML analysis was executed on the CIPRES Science Gateway portal (https://www.phylo.org, accessed on 25 October 2023) [26] using RAxML-HPC2 on XSEDE v. 8.2.12 [27,28,29,30]. Bayesian inference was performed on a server with a Linux system. The ML analysis utilized default parameters, while the BI analysis was configured with a quick boot employing an automatic stop option. The Bayesian inference comprised four parallel runs of 50 million generations, employing stopping rules and a sampling frequency of 100 generations. The burn-in score was set to 0.25, and posterior probability (PP) was determined based on the remaining tree. The iTOL website (https://itol.embl.de, accessed on 25 October 2023) was utilized for plotting all the tree results and optimizing the tree construction and layout. The final results were presented using Adobe Illustrator CC 2019.

## 3. Results

### 3.1. Phylogenetic Analyses

A phylogenetic analysis was conducted on a dataset comprising 143 isolates, including 173 strains sourced from GenBank and 6 strains isolated in-house, collectively representing various *Phyllosticta* species. Among these, 141 isolates were categorized as the ingroup, while two strains, *Botryosphaeria stevensii* (CBS 112553) and *Botryosphaeria obtusa* (CMW8232), were employed as outgroup references. The ultimate alignment consisted of 3339 concatenated characters, distributed as follows: 1–726 (ITS), 727–1466 (LSU), 1467–1980 (TEF1α), 1981–2248 (ACT), and 2249–3339 (GPDH). Among these, 1923 characters remained constant, 252 were variable and parsimony-uninformative, and 1164 were parsimony-informative. The maximum likelihood (ML) tree and Bayesian tree exhibited a comparable topological structure, with the Bayesian tree topology being consistent with the ML tree (Figure 1). The phylogenetic analysis based on five genes categorized the 143 strains into 98 species (Figure 1). The loci, including ITS, LSU, TEF1α, and GPDH, were analyzed using the GTR + I + G model, while the ACT locus was analyzed using the GTR + G model. The Markov chain Monte Carlo (MCMC) analysis for the concatenated genes ran for 5,455,000 generations, with the subsequent trees used to calculate posterior probabilities in the majority rule consensus trees. This study revealed three new species, namely *Phyllosticta fujianensis* sp. nov., *P. saprophytica* sp. nov., and *P. turpiniae* sp. nov.

In the phylogenetic analysis conducted in this study, *Phyllosticta saprophytica* (SAUCC1516-2, SAUCC1516-5) was positioned within the *P. cruenta* species complex and exhibited a close relationship with *P. sicmea* (CGMCC3.14354), garnering strong support (1.0 BIPP and 100% MLBV). Within the *P. centrica* species complex, both *P. fujianensis* (SAUCC1366-3, SAUCC1366-6) and *P. turpiniae* were identified. *P. fujianensis* showed an evolutionary relationship with *P. citribraziliensis* (CBS 100098) and *P. ericarum* (CPC19744) in the phylogenetic tree. However, the MLBV support rate was lower (30%), and the BIPP support rate was higher (0.8 BIPP). *P. turpiniae* (SAUCC2864-3 and SAUCC2864-5) exhibited a close relationship with *P. speewahensis* (BRIP 58044), receiving robust support (1.0 BIPP and 98% MLBV).

### 3.2. Taxonomy

#### 3.2.1. *Phyllosticta fujianensis* Y. Jiang, Z.X. Zhang & X.G. Zhang, sp. nov.

MycoBank—No. MB850038.

Etymology—The specific epithet “*fujianensis*” refers to Fujian City (China) where the type was collected.

Diagnosis—*Phyllosticta fujianensis* can be distinguished from phylogenetically similar *P. citribraziliensis* based on its shorter and narrower conidia (10.0–14.0 × 1.5–2.0 µm *P. fujianensis* vs. 7.0–20.0 × 3.0–4.0 µm in *P. citribraziliensis*). *P. fujianensis* differs from *P. citribraziliensis* by 58 nucleotides (8/620 in ITS, 5/1241 in LSU, 38/386 in TEF1α, 7/217 in ACT). *P. fujianensis* can be distinguished from *P. ericarum* based on its narrower conidia (10.0–14.0 × 1.5–2.0 µm *P. fujianensis* vs. 10.0–12.0 × 6.0–7.0 µm *P. ericarum*). *P. fujianensis* differs from *P. centrica* by 36 nucleotides (6/620 in ITS, 24/404 in TEF1α and 6/217 ACT).

Type—China, Fujian Province: Wuyishan National Park, found on diseased leaves of *Lonicera japonica*, 15 October 2022, Y. Jiang, holotype HMAS 352648, ex-holotype living culture SAUCC 1366-3.

Description—*Phyllosticta fujianensis* is leaf endogenic and associated with leaf spots. Its asexual morph exhibits pycnidial conidiomata, which are mostly aggregated in clusters, black, and erumpent. In PDA culture, it exudes write conidial masses within 10 days or longer. The conidial peduncle is not obvious and usually degenerated into a conidial cell. The conidiogenous cells terminal is 10.0–14.0 × 1.5–2.0 μm and subcylindrical, ampulliform, hyaline, and smooth. The conidia hyaline is 10.0–11.0 × 3.5–5.0 μm, mean ± SD = 10.7 ± 0.4 × 5.0 ± 0.2 μm, without a diaphragm, with thin, smooth walls. It is ovoid, ampoule-shaped, elliptic, or nearly spherical, wrapped in a thin mucous sheath of 1.0–2.0 μm, and surrounded by hyalurons. The tip has a slimy appendage, and it is unbranched, soft, and tapering toward the tip. The sexual morphs are unknown; see Figure 2.

Culture characteristics—After 14 days at 25 °C in the dark, colonies of 65–76 mm diameter were found on the PDA, growing at 4.6–5.4 mm/day. The colonies were greenish–black on the front and back sides, with moderate aerial mycelium on the surface and black social conidia.

Additional specimen examined—China, Fujian Province: Wuyishan National Park, found on diseased leaves of *Lonicera japonica*, 15 October 2022, Z.X. Zhang, HSAUP 1366-6; living culture SAUCC 1366-6.

Notes—Phylogenetic analysis of five genes indicates that *Phyllosticta fujianensis* belongs to the *P. centrica* species complex and is closely related to *P. citribraziliensis* (CBS 100098) and *P. ericarum* (CBS 132534) in phylogeny (Figure 1). However, *P. fujianensis* differs from *P. centrica* by 36 nucleotides (6/620 in ITS, 24/404 in TEF1α and 6/217 ACT) and from *P. citribraziliensis* by 58 nucleotides (8/620 in ITS, 5/1241 in LSU, 38/386 in TEF1α, 7/217 in ACT). Morphologically, *P. fujianensis* can be distinguished from *P. citribraziliensis* based on its shorter and narrower conidia (10.0–14.0 × 1.5–2.0 µm *P. fujianensis* vs. 7–20 × 3–4 µm in *P. citribraziliensis*) [31]. *P. ericarum* can be distinguished from *P. ericarum* based on its narrower conidia (10.0–14.0 × 1.5–2.0 µm *P. fujianensis* vs. 10–12 × 6–7 µm *P. ericarum*) [32] Therefore, the species was considered as a new species based on its morphology as well as a phylogenetic analysis of five gene loci. For details, see Table 2.

#### 3.2.2. *Phyllosticta saprophytica* Y. Jiang, Z.X. Zhang & X.G. Zhang, sp. nov.

MycoBank—No. MB850234.

Etymology—The specific epithet “*saprophytica*” refers to the host plant’s saprophytic leaves.

Diagnosis—*Phyllosticta saprophytica* can be distinguished from *P. saprophytica* phylogenetically. It differs from *P. schimae* by having shorter conidiogenous cells (11.0–15.0 × 2.5–3.5 μm vs. 8.0–30.0 × 2.0–4.0 μm) and several loci (1/535 in ITS, 8/1227 in LSU, 10/379 in TEF1α, 1/231 in ACT, 1/623 in GPDH).

Type—China, Fujian Province: Wuyishan National Forest Park, found on diseased leaves of saprophytic leaves, 15 October 2022, Y. Jiang, holotype, HMAS 352649, ex-holotype living culture SAUCC 1516-2.

Description—The saprophytic leaf is endogenic and associated with saprophytic leaf spots. The asexual morph exhibits pycnidial conidiomata, mostly clustered and black, appearing erumpent. The PDA cultures oozed black conidial clumps, and pycnidia appeared after 10 days or more. The terminal ends of conidiogenous cells are oval to ampule-shaped, smooth, and clear, measuring 11.0–15.0 × 2.5–3.5 μm. The conidia measure 13.0–15.0 × 5.5–7.0 μm, mean ± SD = 13.4 ± 1.2 × 5.8 ± 0.3 μm. The conidia are solitary, transparent, without a diaphragm, with thin, smooth walls. They exhibit rough, oily spots or large central spots and are ovoid, ampulliform, oval or approximately spherical in shape. They are surrounded by a thin mucus sheath with a thickness of 1.3–2.7μm and are transparent at the top of the attachments, measuring 3.0–8.5 × 1.0–1.5 μm. The tip has a slimy appendage, is unbranched, soft, and tapers toward the tip. The sexual morphs unknown; see Figure 3.

Culture characteristics—The colony on PDA grew on the whole 90 mm petri dish at 25 °C for 14 days, with a growth rate of 6.0–6.5 mm/day. The front and back sides of the colony were black–green, there was a medium number of air mycelia on the surface, and there were black conidia gathered together.

Additional specimen examined—China, Fujian Province: Wuyishan National Forest Park, found on diseased Saprophytic leaves, 15 October 2022, Z.X. Zhang, HSAUP 1516-5, living culture, SAUCC 1516-5.

Notes—In the phylogeny analyses, *Phyllosticta saprophytica* forms a sister group with *P. schimae* (CGMCC 3.14354). When comparing DNA sequences between *P. saprophytica* and *P. schimae* (CGMCC 3.14354), a distinction is evident, with *P*. *saprophytica* differing from *P*. *schimae* by 21 nucleotides (1/535 in ITS, 8/1227 in LSU, 10/379 in TEF1α, 1/231 in ACT, 1/623 in GPDH). Furthermore, *P. saprophytica* exhibited shorter conidiogenous cells compared to *P. schimae* (11.0–15.0 × 2.5–3.5 μm vs 8.0–30.0 × 2.0–4.0 μm) [33]. Therefore, we established this strain as *P. saprophytica*. For details, see Table 2.

#### 3.2.3. *Phyllosticta turpiniae* Y. Jiang, Z.X. Zhang & X.G. Zhang, sp. nov.

MycoBank No. MB850290.

Etymology—The specific epithet “*turpiniae*” refers to the leaves of the host plant, *Turpinia arguta*.

Diagnosis—*Phyllosticta turpiniae* can be distinguished from *P. hostea* by the presence of longer and wider conidia (17.0–22.0 × 8.0–10.0 µm in *P. turpiniae* vs. 8.0–15.0 × 5.0–9.0 µm in *P. hostea*). When comparing the DNA sequences of *P. turpiniae* with *P. speewahensis*, there is a 92.9% (576/620 identities; 11/620 gaps) sequence similarity in ITS, a 98.4% (817/830 identities, 0/830 gaps) similarity in LSU, a 94.4% (322/341 identities, 4/341 gaps) similarity in TEF1α, and a 91.8% (161/176 identities, 7/176 gaps) similarity in ACT.

Type—China, Fujian Province: Wuyishan National Forest Park, found on diseased leaves of *Turpinia arguta*, 15 October 2022, Y. Jiang, holotype, HMAS 352650, ex-holotype living culture SAUCC 2864-3.

Description—Originating from endogenous leaves and related to leaf spots, the asexual morph of this species features pycnidial conidiomata, mostly agglomerating into black clumps. On the PDA medium, ivory conidia were exuded for 10 days or more. The conidiogenous cells are terminal, subcylindrical, ampulliform, hyaline, and smooth, measuring 9.0–20.0 × 2.5–5.0 μm. The conidia are 17.0–22.0 × 8.0–10.0 μm, mean ± SD = 19.8 ± 1.8 × 9.5 ± 0.5 μm. The conidia are hyaline, aseptate, thin, and smooth walled, coarsely guttulate or with a single large central guttule. They are ovoid, ampulliform, ellipsoidal to subglobose, and enclosed in a thin mucoid sheath measuring 1.0–2.0 μm thick. The conidia also bear a hyaline, apical mucoid appendage, measuring 3.0–8.5 × 1.0–1.5 μm. The appendage is flexible, unbranched, and tapers towards an acutely rounded tip. The sexual morphs unknown; see Figure 4.

Culture characteristics—The colonies growing on PDA were cultured in darkness at 25 °C for 14 days, and the colonies grew in 90 mm petri dishes to 60–80 mm, with a growth rate of 4.2–5.7 mm/day. The front and back were black–green or black. There was air mycelium on the surface, and the black conidia were clustered together.

Additional specimen examined—China, Fujian Province: Wuyishan National Forest Park, found on diseased leaves of *Turpinia arguta*, 15 October 2022, Z.X. Zhang, HSAUP 2864-5; living culture SAUCC 2864-5.

Notes—In the phylogeny analyses, *Phyllosticta turpiniae* forms a sister group with *P. hostea* (BRIP 58044). A comparison of the DNA sequences of *P. turpiniae* with *P. speewahensis* (CGMCC 3.14355) revealed that there is a 92.9% (576/620 identities; 11/620 gaps) sequence similarity in ITS, 98.4% (817/830 identities, 0/830 gaps) similarity in LSU, 94.4% (322/341 identities, 4/341 gaps) similarity in TEF1α, and 91.8% (161/176 identities, 7/176 gaps) similarity in ACT. Morphologically, *P. turpiniae* can be distinguished from *P. hostea* by having longer and wider conidia (17.0–22.0 × 8.0–10.0 µm in *P. turpiniae* vs. 8.0–15.0 × 5.0–9.0 µm in *P. hostea*) [33]. Therefore, the species was considered as a new species based on its morphology as well as a phylogenetic analysis of five gene loci. For details, see Table 2.

## 4. Discussion

The three isolates of *Phyllotomycetes* were obtained from Wuyishan National Park, Fujian Province, China, situated at coordinates 117°–118° E and 27°–28° N. This region experiences a typical subtropical monsoon climate, characterized by warm and humid conditions accompanied by ample rainfall. This climatic environment is conducive to the thriving growth of diverse microorganisms. *Phyllosticta* species identification traditionally relies on a combination of morphological characteristics and host association. However, due to the challenge of similar morphological features for taxonomic identification and homology analysis, the classification of *Phyllosticta* species has been complex, leading to a cluttered taxonomy. With advancements in molecular biology, the application of molecular data for species phylogeny has become increasingly sophisticated [5,33,43,52]. Wang et al. [8] introduced six species complexes in *Phyllosticta* based on six gene loci, including the internal transcribed spacer of ribosomal RNA (ITS rDNA), large subunit of ribosomal RNA (LSU rDNA), translation elongation factor 1 alpha (TEF1α), actin (ACT), and glycerol-3-phosphate dehydrogenase (GPDH). The ITS, along with other loci such as LSU, TEF1α, ACT, and GPDH, allows for phylogenetic identification at the species level [5,53]. In the present study, phylogenetic analyses of *Phyllosticta* species, as accepted in the latest paper, were conducted based on five loci. In this study, phylogenetic analyses of *Phyllosticta* species, as acknowledged in the most recent literature, were undertaken based on five genetic loci. The three newly discovered species were found within the *P. capitalensis* species complex and *P. concentrica* species complex. Consequently, the primary emphasis of the investigation was directed towards understanding the intricacies of the *P. capitalensis* and *P. concentrica* species complexes.

In this paper, we present a comprehensive examination of the multilocus phylogeny involving three *Phyllosticta* species isolates sourced from two host genera and saprophytic leaves. Additionally, we provide detailed illustrations of the morphological characteristics observed in culture. These isolates contribute novel insights into the species diversity of Phyllosticta within Fujian, China. We propose the recognition of three new species: *Phyllosticta fujianensis*, *Phyllosticta saprophytica*, and *Phyllosticta turpiniae*. *Phyllosticta fujianensis* was isolated from *Lonicera japonica* in Fujian Province, *P. saprophytica* was obtained from saprophytic leaves in Fujian Province, and *P. turpiniae* was isolated from *Turpinia montana* in Fujian Province. *Phyllosticta* exhibits typical morphological features, with asexual conidia being oval or oval to obovate, pearly-shaped, and featuring a mucous sheath and apical mucous appendage. The isolated strains conform to the morphological characteristics of *Phyllosticta*.

As of 25 October 2023, the Global Biodiversity Information Facility (GBIF) (https://www.gbif.org/, accessed on 25 October 2023) encompasses 9678 geo-referenced records of *Phyllosticta* species reported globally. The main distribution locations for these species are in America, Asia, and Europe, with the United States having the most extensive distribution [54,55]. Among the *Phyllosticta* species, *P. carbitalensis*, identified as a relatively weak plant pathogenic agent, is known to induce leaf spot diseases in various plants, including tea (*Camellia sinensis*), oil palm (*Elaeis guineensis*), *Ricinus communis*, and guava black spot [56,57]. *P. concentrica* has been observed to cause leaf spot in *Hedera helix* and *Boehmeria cylindrica*. According to the current study, *Phyllosticta* is extensively distributed in China, and numerous researchers have reported new species of *Phyllosticta* and new records in the country. Contributions from Wang et al., Lin et al., Su et al., Liao et al., Tang et al., and Zhang et al. [8,14,33,44,45,57,58,59,60] have notably enhanced the accuracy of *Phyllosticta* classification. This precision holds significant implications for the utilization and development of *Phyllosticta* species. In summary, *Phyllosticta* species showcase a wide distribution and manifest diverse lifestyles, encompassing pathogenicity, latency, and endophytic characteristics. Further research is warranted to ascertain whether *Phyllosticta* species exhibit host specificity and to elucidate the conditions governing the transition from endophytic to pathogenic behavior.

## Figures and Tables

**Figure 1 jof-10-00007-f001:**
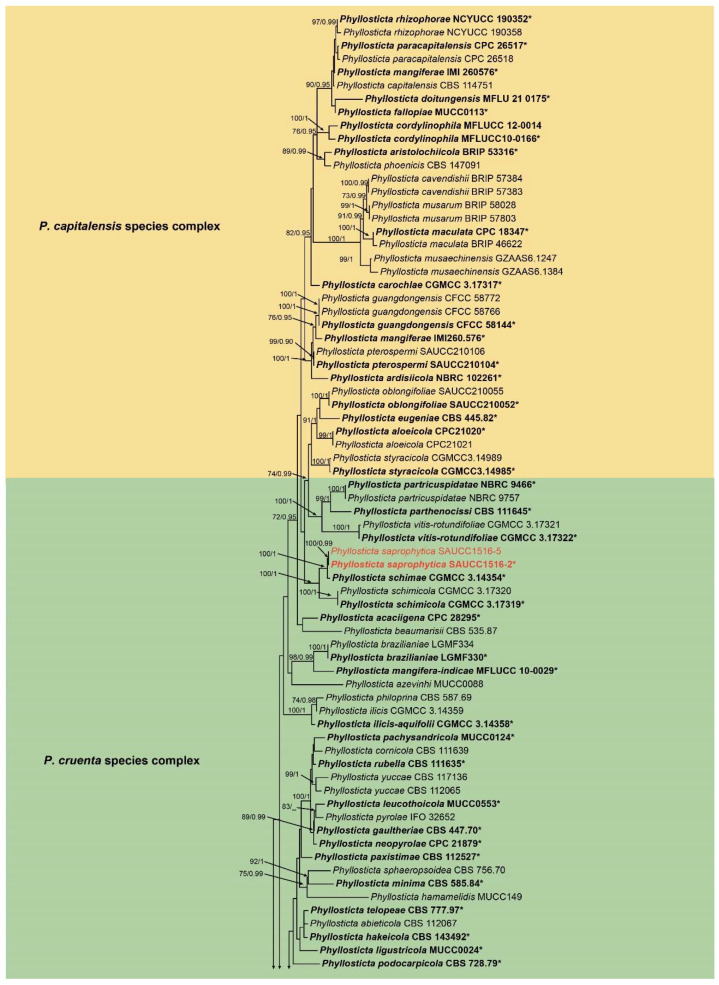
Phylogram of the genus *Phyllosticta* based on a concatenated ITS, LSU, TEF1α, ACT, and GPDH sequence alignment, with *Botryosphaeria obtusa* (CMW 8232) and *Botryosphaeria stevensii* (CBS 112553) serving as outgroups. Maximum likelihood bootstrap support values and Bayesian inference posterior probabilities above 70% and 0.90 are shown at the first and second position, respectively. Ex-type cultures are indicated in bold face. Strains obtained in the current study are in red. Some branches are shortened for layout purposes—these are indicated by two diagonal lines with the number of times. The bar at the bottom-left represents the substitutions per site. Notes: Ex-type or ex-holotype strains are labeled with a star mark “*”.

**Figure 2 jof-10-00007-f002:**
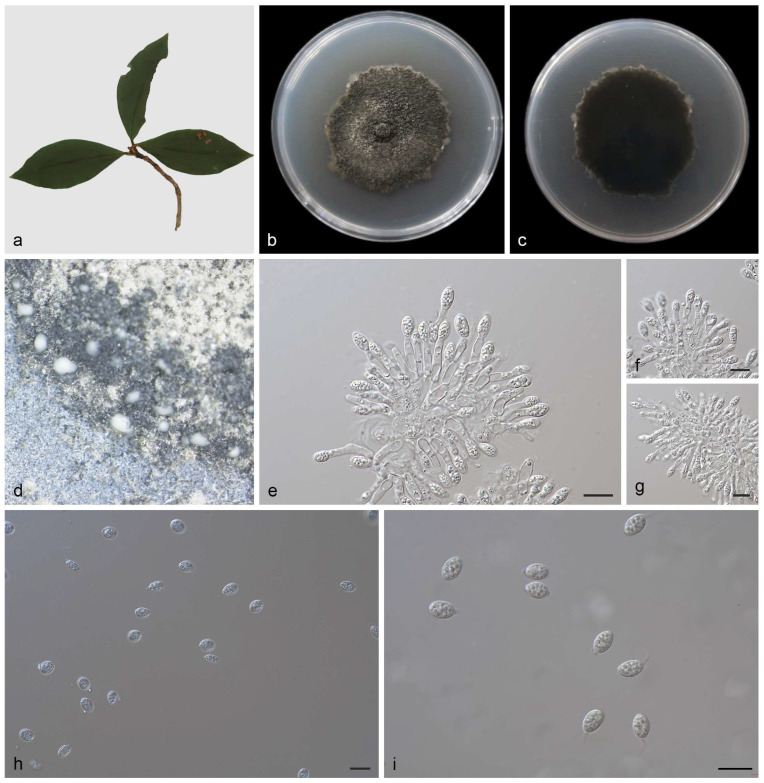
*Phyllosticta fujianensis* (holotype HMAS 352648) (**a**) diseased leaf of *Lonicera japonica* (**b**,**c**) colonies after 15 days on PDA (**d**) conidiomata (**e**–**g**) conidiogenous cells with conidia (**h**,**i**) conidia. Scale bars: 10 μm (**e**–**i**).

**Figure 3 jof-10-00007-f003:**
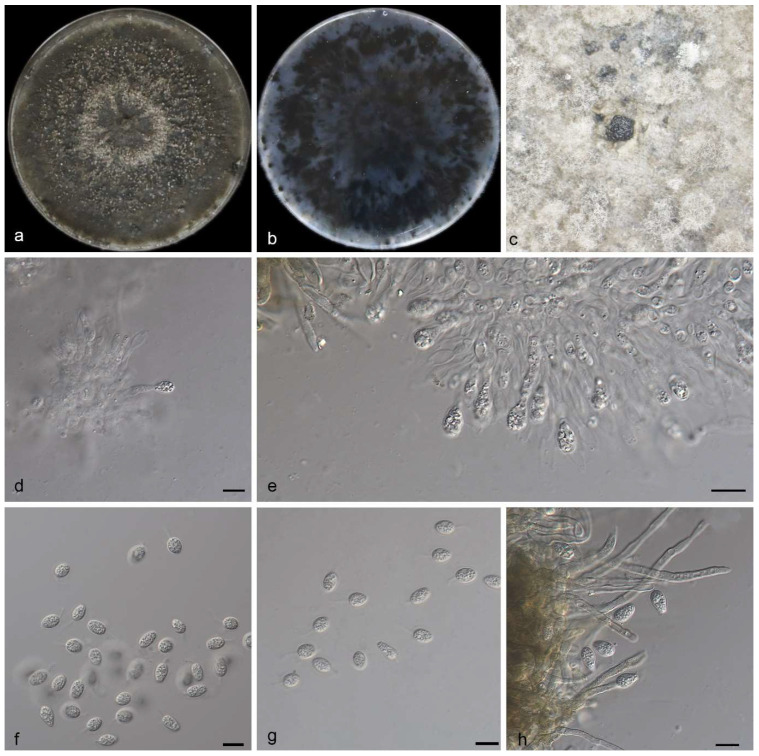
*Phyllosticta saprophytica* (holotype HMAS 352649) (**a**,**b**) colonies after 15 days on PDA. (**c**) Conidiomata, (**d**,**e**) conidiogenous cells with conidia, (**f**–**h**) conidia. Scale bars: 10 μm (**d**–**h**).

**Figure 4 jof-10-00007-f004:**
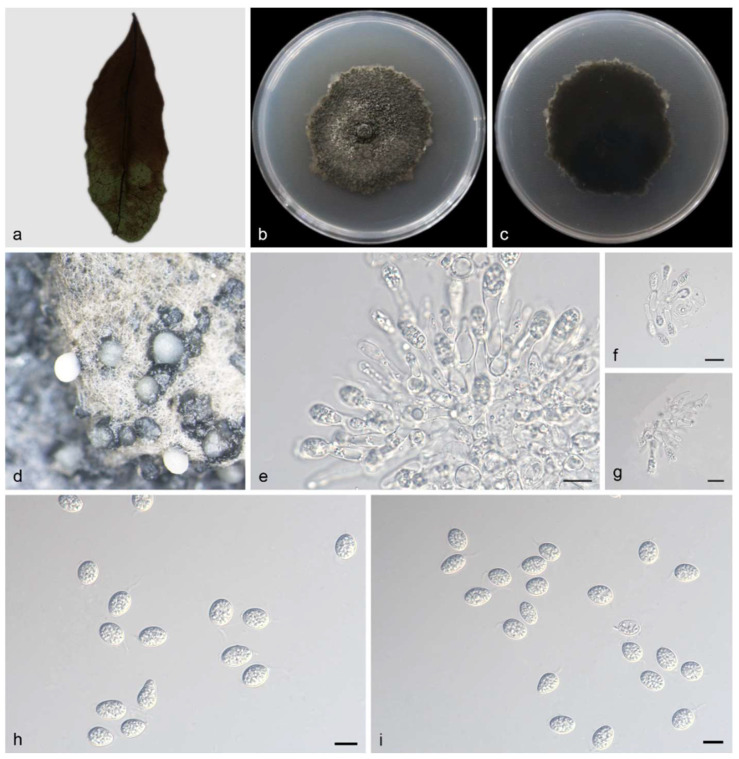
*Phyllosticta turpiniae* (holotype HMAS 352650). (**a**) Diseased leaf of *Turpinia arguta*, (**b**,**c**) colonies after 15 days on PDA, (**d**) conidiomata, (**e**–**g**) conidiogenous cells with conidia, (**h**,**i**) conidia. Scale bars: 10 μm (**e**–**i**).

**Table 1 jof-10-00007-t001:** Molecular markers and their PCR primers and programs used in this study.

Loci	PCR Primers	Sequence (5′–3′)	PCR Cycles	References
ITS	ITS5	GGA AGT AAA AGT CGT AAC AAG G	(94 °C: 30 s, 55 °C: 30 s, 72 °C: 1 min) × 35 cycles	[16]
	ITS4	TCC TCC GCT TAT TGA TAT GC		
LSU	LR0R	GTA CCC GCT GAA CTT AAG C	(94 °C: 30 s, 51 °C: 30 s, 72 °C: 1 min) × 35 cycles	[17,18]
	LR7	TAC TAC CAC CAA GAT CT		
TEF1α	EF1-728F	CAT CGA GAA GTT CGA GAA GG	(94 °C: 30 s, 48 °C: 30 s, 72 °C: 1 min) × 35 cycles	[19,20]
	EF2	GGA RGT ACC AGT SAT CAT GTT		
ACT	ACT-512F	ATG TGC AAG GCC GGT TTC GC	(94 °C: 30 s, 52 °C: 30 s, 72 °C: 1 min) × 35 cycles	[20]
	ACT-783R	TAC GAG TCC TTC TGG CCC AT		
GPDH	Gpd1-LM	ATT GGC CGC ATC GTC TTC CGC AA	(94 °C: 30 s, 52 °C: 30 s, 72 °C: 1 min) × 35 cycles	[21]
	Gpd2-LM	CCC ACT CGT TGT CGT ACC A		

**Table 2 jof-10-00007-t002:** The asexual morphological characters of some *Phyllosticta* species.

Species	Conidiogenous Cells	Size of Conidiogenous Cells (μm)	Conidia	Size of Conidia (μm)	References
*Phyllosticta rhizophorae*	Reduced to conidiogenous cells, subcylindrical to ampulliform	13.0–25.0 × 3.0–5.0 μm	Terminal, subcylindrical, hyaline, smooth	10.0–17.0 × 3.0–5.0 μm	[5]
* P. oblongifoliae *	Indistinct	9.0–14.0 × 2.5–4.5 μm	Hyaline, aseptate, ovoid, ampulliform, ellipsoidal to subglobose	8.0–13.0 × 6.0–8.0 μm	[7]
* P. pterospermi *	Cylindrical, hyaline, smooth	7.5–11.0 × 2.5–4.5 μm	Hyaline, aseptate, thin- and smooth-walled, obovoid, ellipsoidal to subglobose	8.0–12.0 × 4.5–8.5 μm	[7]
* P. anhuiensis *	Phialidic, hyaline, thin-walled, smooth, subcylindrical to ampulliform	10.0–16.0 × 2.5–4.5 μm	Solitary, hyaline, aseptate, thin- and smooth-walled, coarsely guttulate, globose or ellipsoid to obvoid	8.5–12.0×5.5–9.0 μm	[8]
* P. guangdongensis *	Subcylindrical to ampulliform, hyaline, smooth	10.0–15.0 × 2.5–4.0 μm	Solitary, hyaline, aseptate, thin and smooth-walled, ellipsoid to obovoid	10.0–14.0 × 6.0–8.0 μm	[8]
* P. musarum *	Cylindrical or conical	4.0–11.0 × 2.5–5.0 µm	One-celled, obovoidal, ellipsoidal or short cylindrical, pyriform when young	15.0–18.0 × 9.0–10.0 µm	[9]
* P. phoenicis *	Subcylindrical to ampulliform, terminal, hyaline, smooth, coated in a mucoid layer	9.0–15.0 × 3.0–4.0 μm	Solitary, hyaline, aseptate, thin- and smooth-walled, granular	10.0–15.0 × 7.0–9.0 μm	[13]
* P. doitungensis *	Terminal, subcylindrical, hyaline, smooth	6.0–8.0 × 2.0–4.0 μm	Solitary, hyaline, aseptate, thin and smooth-walled, with a single, large central guttulate, tapering	5.0–8.0 × 3.5–6.0 μm	[22]
* P. gwangjuensis *	Subcylindrical, ampulliform, hyaline	8.5–22.5 × 3.5–5.5 μm	Ovoid to ellipsoid shape, rounded at both ends	10.0–13.5 × 7.0–9.0 μm	[23]
* P. citribraziliensis *	Subcylindrical to doliiform, hyaline, smooth, coated in a mucoid layer	7.0–20.0 × 3.0–4.0 μm	Solitary, hyaline, aseptate, thin- and smooth-walled, coarsely guttulate, ellipsoid to obovoid	10.0–12.0 × 6.0–8.0 μm	[31]
* P. bifrenariae *	Subcylindrical to ampulliform, hyaline, smooth	7.0–10.0 × 4.0–5.0 μm	Solitary, hyaline, aseptate, thin- and smooth-walled, ellipsoid to ovoid or obovoid, tapering toward a narrowly truncate base	11.0–13.0 × 8.0–9.0 μm	[31]
* P. ericarum *	Subcylindrical, hyaline, smooth, coated in a mucoid layer	12.0–20.0 × 3.0–4.0 μm	Solitary, hyaline, aseptate, thin- and smooth walled, coarsely guttulate	8.0–12.0 × 6.0–7.0 μm	[32]
* P. hostae *	Holoblastic, phialidic, cylindrical, subcylindrical to ampulliform, hyaline, thin-walled, smooth	7.0–22.0 × 2.0–5.0 μm	Unicellular, thin- and smooth-walled, ellipsoid, subglobose to obovoid, with a large central guttule	8.0–15.0 × 5.0–9.0 μm	[33]
* P. ilicis-aquifolii *	Holoblastic, phialidic cylindrical, subcylindrical to ampulliform, hyaline, thin-walled, smooth	12.0–17.0 × 3.0–4.0 μm,	Unicellular, thin-and smooth-walled, globose ellipsoid to obovoid	12.0–17.0 × 2.0–3.0 μm	[33]
* P. schimae *	Holoblastic, phialidic, short cylindrical, subcylindrical to ampulliform, hyaline, thin-walled, smooth	8.0–30.0 × 2.0–4.0 μm	Unicellular, thin- and smooth-walled, globose, ellipsoid to obovoid	7.0–13.0 × 4.0–7.0 μm	[33]
* P acaciigena *	Terminal, subcylindrical, hyaline, smooth, coated in a mucoid layer	7.0–15.0 × 3.0–5.0 μm	Solitary, hyaline, aseptate, thin and smooth-walled, coarsely guttulate, ellipsoid to obovoid	12.0–15.0 × 7.0–8.0 μm	[34]
* P. ardisiicola *	Holoblasticae, hyalinae, cylindricae vel conicae	5.0–12.5 × 1.2–2.5 μm	Globosa, elliptica vel obovata, primo basi truncata, posterius utrinque rotundata	7.0–11.0 × 5.0–7.5 μm	[35]
* P. aspidistricola *	Hyalinae, cylindricae, conicae vel lageniformes	7.0–12.5 × 1.2–2.5 μm	Globosa, elliptica vel obovata, primo basi truncata, posterius utrinque rotundata	9.5–12.5 × 8.5–10.0 μm	[35]
* P. fallopiae *	Holoblasticae, hyalinae, cylindricae vel conicae	5.0–10.0 × 1.2–2.5 μm	globosa, elliptica vel obovata, primo basi truncata, posterius utrinque rotundata	8.5–12.5 × 6.0–7.5 μm	[35]
* P. aristolochiicola *	Hyaline, smooth, subcylindrical to ampulliform	10.0–20.0 × 2.0–4.0 μm	Subglobose, bovoid, rounded apex, hyaline	7.0–16.0 × 6.5–11.0 μm	[36]
* P. beaumarisii *	Distinctive	Distinctive	One-celled, hyaline, ovoid–ellipsoidal	7.5–15.0 × 6.5–8.7 μm	[37]
* P. carochlae *	Holoblastic, hyaline, cylindrical	2.0–6.5 × 5.5–13.0 μm	Unicellular, ovoid, obovoid, ellipsoidal to subglobose	6.0–8.5 × 10.0–12.0 μm	[38]
* P. partricuspidatae *	Holoblastic, hyaline, long cylindrical	2.0–6.0 × 3.5–12.0 μm	Unicellular, thin- and smooth-walled, ampulliform, ovoid, obovoid, ellipsoidal to subglobose	5.0–8.5 × 8.0–12.0 μm	[38]
* P. schimicola *	Holoblastic, hyaline, long cylindrical, subcylindrical to ampulliform	1.5–4.5 × 5.0–12.0 μm	Unicellular, smooth-walled, ovoid to long ovoid, ampulliform, ellipsoidal to subglobose	5.0–8.0 × 8.0–11.0 μm	[38]
* P. vitis-rotundifoliae *	Holoblastic, hyaline, long cylindrical, subcylindrical to ampulliform	5.0–11.0 × 2.0–5.5 μm	Unicellular, thin- and smooth-walled, ampulliform, ovoid, obovoid, ellipsoidal to subglobose, truncate at the base	6.0–9.5 × 9.0–13.0 μm	[38]
* P. cavendishii *	Doliiform or subcylindrical, solitary, hyaline	8.0–12.0 × 4.0–5.0 μm,	Solitary, hyaline, aseptate, coarsely guttulate, thin and smooth-walled	13.0–16.0 × 8.0–9.0 μm	[39]
*P. maculata*	Subcylindrical to doliiform, solitary, hyaline	9.0–12.0 × 4.0–5.0 μm	Oblong or obovoid to subclavate, apex solitary, hyaline	16.0–19.0 × 10.0–12.0 μm	[39]
* P. eugeniae *	Distinctive	Distinctive	Ovate, hyaline, granular	9.6–16.8 × 4.8–6.0 μm	[40]
* P. kerriae *	Hyalinae, cylindricae vel conicae	5.0–7.5 × 1.2–2.5 μm	Obovata, primo basi truncata, posterius utrinque rotundata	9.5–12.5 × 6.0–7.5 μm	[40]
* P. citrimaxima *	Phialidic, cylindrical, hyaline	3.0–5.0 × 1.0–2.0 μm	Ellipsoidal, hyaline, one-celled, smooth	5.0–8.0 × 3.0–7.0 μm	[41]
* P. mangifera-indicae *	Lining the inner wall, phialidic, cylindrical, hyaline	3.0–5.0 × 3.0–4.0 μm	Ellipsoidal, hyaline, aseptate, smooth	6.0–13.0 × 4.0–6.0 μm	[41]
* P. musaechinensis *	Hyaline, aseptate, coarsely guttulate, ellipsoidal or clavate, thin- and smooth-walled	14.0–18.0 × 8.0–12.0 μm	Hyaline, aseptate, coarsely guttulate, ellipsoidal, clavate or irregular, thin- and smooth-walled	5.5–22.5 × 8.5–13.0 μm	[42]
* P. paracapitalensis *	Terminal, subcylindrical, hyaline, smooth, coated in a mucoid layer	7.0–15.0 × 3.0–4.0 μm	Solitary, hyaline, aseptate, thin and smooth-walled, granular	3.0–14.0 × 6.0–7.0 μm	[43]
* P. paracitricarpa *	Subcylindrical, hyaline, smooth, coated in a mucoid layer	12.0–17.0 × 3.0–4.0 μm	Solitary, hyaline, aseptate, thin- and smooth-walled, granular	11.0–15.0 × 7.0–9.0 μm	[43]
* P. parthenocissi *	Cylindrical or conical	7.5–12.5 × 2.5–3.5 μm	One-celled, globose or subglobose	7.5–10.0 × 6.0–9.0 μm	[44]
* P. styracicola *	Holoblastic, hyaline, cylindrical	2.0–3.5 × 8.0–12.5 μm	One-celled, ellipsoidal to subglobose, surrounded by a thick mucilaginous layer	9.5–13.0 × 6.5–9.0 μm	[45]
* P. catimbauensis *	Terminal, subcylindrical to ampulliform, hyaline, smooth	9.5–10.5 × 3–3.5 μm	Solitary, hyaline, aseptate, thin- and smooth-walled, granular, ellipsoid, globose, subglobose	8.5–10.5 × 5.5–6 μm	[46]
* P. citrichinensis *	Holoblasticae, phialidicae, breviter cylindricae, lageniformes, singulae, hyalina, tenui-muratas, laevia	6.0–12.0 × 2.0–5.0 μm	Elliptica vel ovoidea, utrinque rotundata, singulae, muratis-levi, strato mucoso circumdantia	8.0–12.0 × 6.0–9.0 μm	[47]
* P. elongata *	Distinctive	Distinctive	Anguste obovatis	9.0–14.0 × 5.0–8.0 μm	[48]
* P. hymenocallidicola *	Terminal, subcylindrical to doliiform, hyaline, smooth, coated with a mucoid layer	7.0–15.0 × 3.0–4.0 μm	Solitary, hyaline, aseptate, thin- and smooth-walled, coarsely guttulate, or with large, single, central guttule, ellipsoid to obovoid	8.0–11.0 × 6.0–7.0 μm	[49]
* P. iridigena *	Doliiform, hyaline, smooth, proliferating percurrently at apex	4.0–7.0 × 4.0–6.0 μm	Solitary, ellipsoid to obovoid, aseptate, smooth, hyaline, guttulate, granular	10.0–15.0 × 7.0–9.0 μm	[50]
* P. ophiopogonis *	Holoblastic, determinate, discrete, hyaline, sometimes, rarely integrated	7.0–12.0 × 2.0–4.0 μm	Hyaline, one-celled, coarse-guttulate, smooth-walled, globose, ellipsoidal, clavate or obclavate	10.0–14.0 × 7.0–8.0 μm	[51]
** * P. saprophytica * **	**Oval to ampule, smooth, clear**	**11.0–15.0 × 2.5–3.5 μm**	**Transparent, without diaphragm, with thin, smooth walls, ovoid, ampulliform**	**13.0–15.0 × 5.5–7.0 μm**	**This study**
** * P. fujianensis * **	**Subcylindrical, ampulliform, hyaline, smooth**	**10.0–14.0 × 1.5–2.0 μm**	**Without diaphragm, with thin smooth walls, ovoid, ampoule-shaped**	**10.0–11.0 × 3.5–5.0 μm**	**This study**
** * P. turpiniae * **	**Terminal, subcylindrical, ampulliform, hyaline, smooth**	**9.0–20.0 × 2.5–5.0 μm**	**Hyaline, aseptate, thin- and smooth-walled**	**17.0–22.0 × 8.0–10.0 μm**	**This study**

Notes: the new species information described in this study is marked in bold.

## Data Availability

The sequences from the present study were submitted to the NCBI database (https://www.ncbi.nlm.nih.gov/, accessed on 25 October 2023). The accession numbers are listed in Appendix A.

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
