# Peer review of "Morphological and Phylogenetic Analyses Reveal Three New Species of Phyllosticta (Botryosphaeriales, Phyllostictaceae) in China"

_jof, 2023, doi:10.3390/jof10010007_

Round 1

Reviewer 1 Report

Comments and Suggestions for Authors

The research is very good, the quality in the presentation of some terms must be incresed as well asthe characteristics of conidiogenous cells and conidia in Table 2 should be standardized. 

Review some more name spelling and include taxonomic authorities 

The diagnoses of the three species that are proposed as new are missing 

It would be good to add a conclusion indicating how the state of knowledge of Phyllosticta remains at least in China.

Author Response

Dear Reviewer,

Thank you for your valuable suggestion. In response to these questions, I answer as follows:

  1. The research is very good, the quality in the presentation of some terms must be incresed as well asthe characteristics of conidiogenous cells and conidia in Table 2 should be standardized.

I agree with the above suggestions and revise them in the article.

  1. Review some more name spelling and include taxonomic authorities.

I accepted the suggestions.

  1. The diagnoses of the three species that are proposed as new are missing.

I accepted the suggestions and added in the article.

  1. It would be good to add a conclusion indicating how the state of knowledge of Phyllosticta remains at least in China

I agree with the above suggestions and revise them in the article.

Best wishes,

Yang Jiang

Reviewer 2 Report

Comments and Suggestions for Authors

The manuscript is too hard to read, I hardly managed to arrive to the end of the manuscript! English is often not correct, verbs are often missing and the text is not edited. I try to give some suggestions, I highlight the sentences more difficult to understand but in this form I don’t recommend this manuscript for publication. The quality of photos is poor. The scientific topic described seems interesting and could be considered in another form.

Comments on the Quality of English Language

The manuscript is too hard to read, I hardly managed to arrive to the end of the manuscript! English is often not correct, verbs are often missing and the text is not edited. I try to give some suggestions, I highlight the sentences more difficult to understand but in this form I don’t recommend this manuscript for publication. The quality of photos is poor. The scientific topic described seems interesting and could be considered in another form.

Author Response

Dear Reviewer,

Thank you for your valuable suggestion. In response to these questions, I answer as follows:

  1. The manuscript is too hard to read, I hardly managed to arrive to the end of the manuscript! English is often not correct, verbs are often missing and the text is not edited. I try to give some suggestions, I highlight the sentences more difficult to understand but in this form I don’t recommend this manuscript for publication. The quality of photos is poor. The scientific topic described seems interesting and could be considered in another form.

I agree with the above suggestions and revise them in the article.

Best wishes,

Yang Jiang

Reviewer 3 Report

Comments and Suggestions for Authors

The authors carried out an unpublished research, where three species of Phyllosticta are identified, in a taxonomic approach evaluating the morphological markers and multicolocus analyses. Although the manuscript is poorly substantiated, it has some doubts that I suggest the authors try to improve.

1. Or am I summarizing this poorly, because it does not include some information on the maorphology of new species, for example is there any morphological marker that can distinguish the described species from other species of Phylloctita?

2. The authors mention two names in the introduction of the sexless (Phylloctita) and sexed (Guignardia) phases, how is the taxonomy, just 1 name? Look for current literature to answer this questão.

Figure 1: I am concerned about the grouping of two isolates SAUCC1516-5 and SAUCC1512-2, it is really different from the species Phylloctita schimae... Credit to the authors who affirm that it is a new species, should be able to validate the nucleotide differences of both "species" .

Figure 2: Better symptoms in the figure, or would the fungus be an endophyte?

Throughout the text, the authors mention the presence of pycnidia, however, they do not show the images.... I recommend that you include images of pycnidia, or the evanescent structures?

Despite these suggestions, I consider the manuscript to be well organized.

Comments on the Quality of English Language

Good

Author Response

Dear Reviewer,

Thank you for your valuable suggestion. In response to these questions, I answer as follows:

  1. Or am I summarizing this poorly, because it does not include some information on the maorphology of new species, for example is there any morphological marker that can distinguish the described species from other species of Phylloctita?

I accepted the suggestions. The article has been revised to add a "Diagnosis" section.

  1. The authors mention two names in the introduction of the sexless (Phylloctita) and sexed (Guignardia) phases, how is the taxonomy, just 1 name? Look for current literature to answer this questão.

“One Fungus = One Name” was proposed in The Amsterdam Declaration on Fungal Nomenclature in 2011,but the genus name of Phyllosticta was retained as the correct genus name.

  1. Figure 1: I am concerned about the grouping of two isolates SAUCC1516-5 and SAUCC1512-2, it is really different from the species Phylloctita schimae... Credit to the authors who affirm that it is a new species, should be able to validate the nucleotide differences of both "species".

I agree with the above suggestions and revise them in the article.

  1. Figure 2: Better symptoms in the figure, or would the fungus be an endophyte.

The isolated strains in Figure 2 are endophytes.

  1. Throughout the text, the authors mention the presence of pycnidia, however, they do not show the images.... I recommend that you include images of pycnidia, or the evanescent structures?

"pycnidia" is a macroscopic image observed under a stereo, and no microscopic images were taken.

Best wishes,

Yang Jiang

Round 2

Reviewer 2 Report

Comments and Suggestions for Authors

I must say, with regret, that the revised version of the manuscript has not changed so much from the original one. The scientific topic is interesting, but the English language is not good enough for a scientific paper. Some little improvements have been made, but I hope I was clear, the text needs a supervision by an English expert, I don’t feel qualified to do this.

Beyond this, revision was not accurate, some mistakes that I reported have not been corrected (like the one in line 57 for example) and indeed I don’t feel like to make the work again; some phrases are non-sense as if they have not been read it accurately.

The suggestion is to accurately revised the manuscript for the English form.

Comments on the Quality of English Language

I must say, with regret, that the revised version of the manuscript has not changed so much from the original one. The scientific topic is interesting, but the English language is not good enough for a scientific paper. Some little improvements have been made, but I hope I was clear, the text needs a supervision by an English expert, I don’t feel qualified to do this.

Beyond this, revision was not accurate, some mistakes that I reported have not been corrected (like the one in line 57 for example) and indeed I don’t feel like to make the work again; some phrases are non-sense as if they have not been read it accurately.

The suggestion is to accurately revised the manuscript for the English form.

Author Response

Dear Reviewer,

Thank you for your valuable suggestion. In response to these questions, I answer as follows:

  1. Beyond this, revision was not accurate, some mistakes that I reported have not been corrected (like the one in line 57 for example) and indeed I don’t feel like to make the work again; some phrases are non-sense as if they have not been read it accurately.

I agree with the above suggestions and revise it in the article.

  1. The suggestion is to accurately revised the manuscript for the English form.

I agree with the above suggestions and revise them in the article.

Best wishes,

Yang Jiang
